# Cement substitution with secondary materials can reduce annual global $CO_2$ emissions by up to 1.3 gigatons

Izhar Hussain Shah [1], Sabbie A. Miller [2], Daqian Jiang[3] & Rupert J. Myers [1]✉

Population and development megatrends will drive growth in cement production, which is already one of the most challenging-to-mitigate sources of $CO_2$ emissions. However, availabilities of conventional secondary cementitious materials (CMs) like fly ash are declining. Here, we present detailed generation rates of secondary CMs worldwide between 2002 and 2018, showing the potential for 3.5 Gt to be generated in 2018. Maximal substitution of Portland cement clinker with these materials could have avoided up to 1.3 Gt $CO_2$-eq. emissions (~44% of cement production and ~2.8% of anthropogenic $CO_2$-eq. emissions) in 2018. We also show that nearly all of the highest cement producing nations can locally generate and use secondary CMs to substitute up to 50% domestic Portland cement clinker, with many countries able to potentially substitute 100% Portland cement clinker. Our results highlight the importance of pursuing regionally optimized CM mix designs and systemic approaches to decarbonizing the global CMs cycle.

Of all the materials used today, cement ranks among the most important—a status that will likely remain so in the future, driven by future trends in development, urbanization, and population growth[1]. The main constituent of cement is Portland cement clinker (hereafter clinker), which is the highly reactive material produced in cement kilns. Clinker is always combined with other constituents in cement, most importantly calcium sulfate to control its reactivity. When cement is mixed with water, a binder forms that is the key glue-like substance in concrete and mortar. Concrete and mortar are used worldwide in buildings and infrastructure, which altogether embed ~46% of all materials extracted from the Earth[2,3]. This massive scale of demand drives ~4 Gt year$^{-1}$ of cement production[4], which is responsible for 7–8% of all anthropogenic $CO_2$ emissions[5]. $CO_2$ emissions from cement production are the third largest source of difficult-to-eliminate emissions, after load-following electricity and iron and steel[6]. Beyond greenhouse gas (GHG) emissions, the production of concrete and mortar cause over ~3% of global energy demand[7], over 5% of global anthropogenic $PM_{10}$ emissions[8], and ~2% of global water withdrawals[9]. These environmental impacts may be reduced through various technical (energy, emissions, and material efficiency) measures, of which cementitious materials (CMs) substitution (including complete and partial substitution) is one of the most promising[10].

Substitution of cement constituents (e.g., clinker), cement, binder, and concrete/mortar are possible. However, substitution in the binder (including cement and its constituents) (Fig. 1) is especially important since: (a) most of the environmental impact in the CMs cycle arises from clinker production[10]; (b) it provides many opportunities for environmentally, technically, and economically beneficial treatment of industrial by-products[11]; and (c) extensive substitution of cement with non-cementitious materials such as steel, bricks, timber, etc., is unlikely in the foreseeable future given the huge global scale needed. For instance, steel and bricks have higher greenhouse gas emissions per unit mass than concrete[12,13] while for timber, a massive expansion of production[3] is needed to achieve comparable substitution rates of timber for concrete[14] to those for clinker substitution in cement (~25 mass%[15]). Concerns related to sustainable forest management provide another barrier to the expansion of timber use. We thus focus on material substitution in cementitious binders here.

[1]Department of Civil and Environmental Engineering, Imperial College London, London, UK. [2]Department of Civil and Environmental Engineering, University of California, Davis, CA, USA. [3]Department of Civil, Construction, and Environmental Engineering, University of Alabama, Tuscaloosa, AL, USA. ✉e-mail: r.myers@imperial.ac.uk

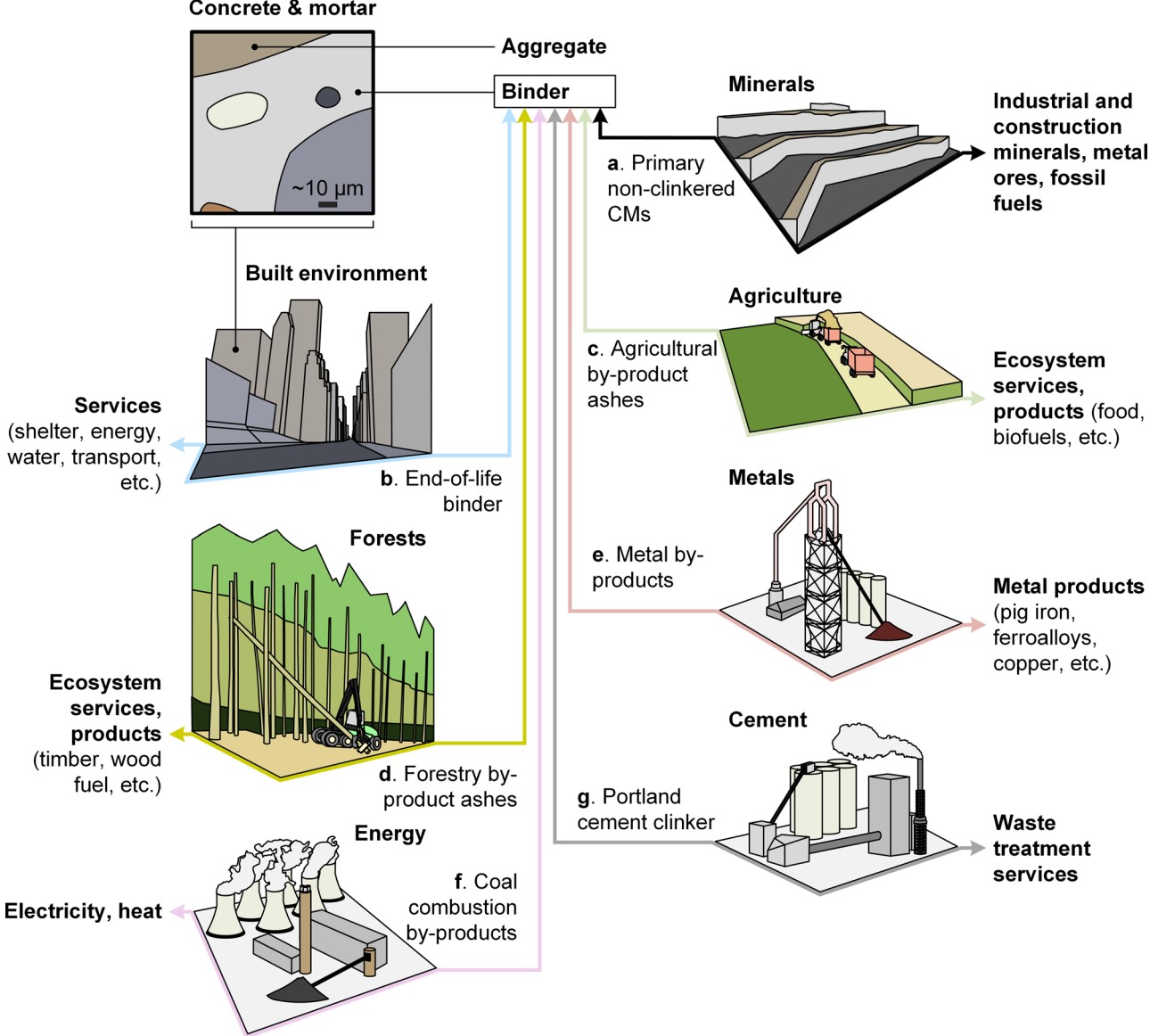

**Fig. 1 | Conceptual representation of clinker and cement substitution.** Substitution in the cementitious binder, and in mortar and concrete (product scale, top left), is linked in several important ways to the wider industrial system (market scale, **a**–**g**). **g** Clinker can be substituted by secondary cementitious materials (CMs) from the (**a**) minerals, (**b**) built environment, (**c**) agriculture, (**d**) forestry, (**e**) metals, and (**f**) energy sectors.

While many substitution cases involving up to ~100% replacement of clinker are possible at the product scale (i.e., 1 kg of cement)[16,17], the potentials of these technologies to reduce environmental impacts at the market scale (i.e., national and global, industry wide) are unclear. Clinker substitution rates may thus be increased through greater use of primary and secondary CMs. Primary CMs such as limestone and kaolinitic clays are globally abundant, and these materials are internationally standardized in cements at up to 35 and 55 mass% respectively[18]. However, the potential supply of secondary CMs that can substitute clinker in cement or binder (Fig. 1) is not systematically reported. This lack of reported knowledge stifles the development and adoption of technology that can realistically lead to extensive clinker substitution at the market scale. It is arguably a key reason why the clinker-to-cement mass ratio has remained at ~0.75 since 2012[15] and mainly limited to a few well-known secondary CMs such as granulated blast furnace slag and coal fly ash, despite the myriad of possibilities.

Here, we systematically and quantitatively review the potential global supply of secondary CMs and their GHG emission reduction benefits. Our analysis covers countries responsible for ~70% of global cement production, and includes key secondary CMs that are among the most widely available and thus have higher potentials to be adopted at scale. We aim to provide key data and the knowledge basis, here and through comprehensive Supplementary Information files, that are needed to guide the development of cement technology towards lower clinker content and achieve related benefits from increased treatment of solid by-products as well as reduced climate change impacts.

## Results

Cement production has more than doubled over the last two decades, from 1.80 Gt in 2002 to 4.05 Gt in 2018 (Fig. 2), due mainly to socio-economic development in China (0.64 Gt in 2002 to 2.2 Gt in 2018)[4]. This rate of increase has occurred faster than the growth in total secondary CMs generation, which fell from 97 mass% (i.e., 1.74 Gt) to 86 mass% (i.e., 3.48 Gt) of cement production between 2002 and 2018, mostly because the combined generation of the two main secondary CMs (i.e., coal fly ash and granulated blast furnace slag) decreased from 25 mass% (i.e., 0.44 Gt) to 17 mass% (i.e., 0.70 Gt) of cement

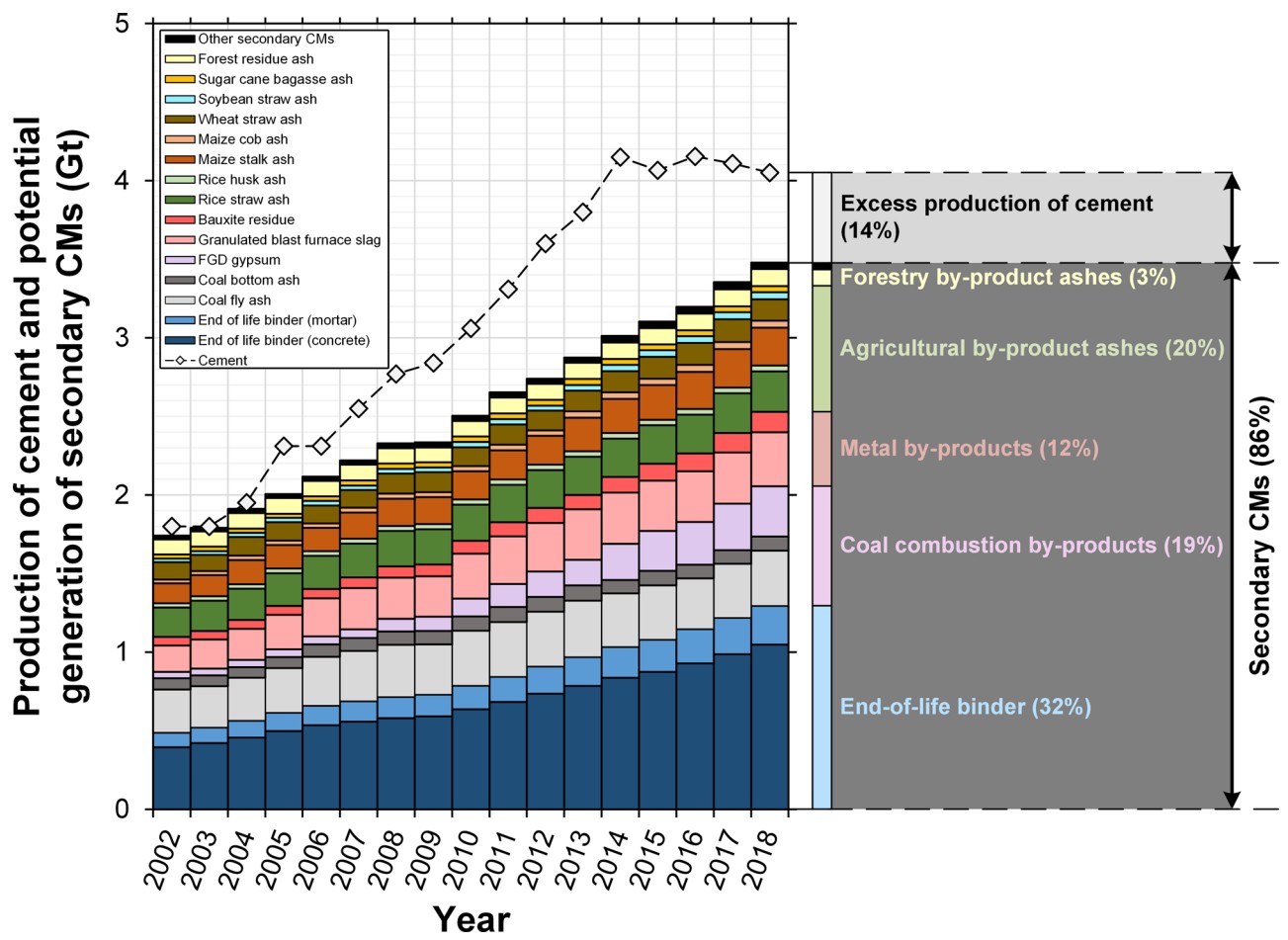

**Fig. 2 | Global production of cement and generation of secondary cementitious materials (CMs).** Cement production data are from[4] and secondary CMs data are modeled here (see Methods).

production over this period (Fig. 2). Therefore, the observed stable ~0.75 clinker-to-cement mass ratio between 2002 and 2018 demonstrates more complete use of coal fly ash and increasing use of other CMs to substitute clinker, notably primary CMs such as limestone[15].

Recently, interest in utilizing primary CMs, notably kaolinitic clays, has increased[19]. However, environmental benefits from their use as cement substitutes are limited in at least two significant ways: (i) they require processing, usually heat treatment that also contributes to environmental impacts, to achieve desirable reactivity in cement systems; and (ii) the extent of substitution is limited by the overall workability (i.e., consistency and cohesiveness of fresh mortar/concrete slurry) and chemistry (i.e., reactivity and solid phase formation) of the cement system. An example is the production of metakaolin from kaolin at 700–850 °C and its subsequent use with limestone to achieve an overall clinker substitution level of ~50 mass%[19]. Here, the heat processing step leads to this material having similar or slightly lower GHG emissions from production to a conventional blended cement with ~30 mass% clinker substitution[20].

On the other hand, secondary CMs, while also limited by the overall workability and chemistry of the cement system, frequently have small environmental impacts relative to clinker production, and are often considered to be burden free in cement LCA studies[16,21–25]. This is because they are usually reactive in their existing forms due to high temperature upstream processing conditions[26] and provide lower revenues than their corresponding main co-products (e.g., pig iron is the main co-product of blast furnace slag). Secondary CMs also span a range of suitable chemistries for cement systems[11]. These desirable factors can be exploited to increase clinker substitution rates well

beyond the historical clinker-to-cement mass ratio (~0.75) through concurrent use of multiple secondary CMs (e.g., standard CEM II, IV, and V cements).

Our data (Fig. 2) show that such secondary CM mixtures can theoretically achieve an average clinker-to-cement mass ratio of ~0.14 globally, i.e., a reduction of ~61 mass% clinker in cement, provided the resulting binders can be used to produce concretes and mortars with appropriate properties (e.g., compressive strength development; see Methods). Various cements with clinker-to-cement mass ratios of ~0.5 are standardized and/or available (e.g., CEM III–V, LC[3,19]). However, progressively decreasing the clinker-to-cement mass ratio is increasingly difficult without alkali-activation, which describes the use of aqueous alkaline solutions (e.g., NaOH (aq), Na₂SiO₃ (aq)) rather than water in the binder formulation[27] (hereafter mix). Alkali-activated materials can be produced without clinker, although production rates of alkaline activators would need to be greatly increased to facilitate this substitution of conventional Portland cement binders at the industrial scale. This requirement is a major barrier to adoption especially considering the demand for such alkaline materials in other industries. For example, the global production rate of NaOH (s) in 2013 was ~80 Mt/year,[28] which is small relative to the amount needed (~300 Mt/year) to maximize utilization of secondary CMs in alkali-activated materials.

In practice, transportation costs are a key limitation in sourcing raw materials for cement production[29]. Preferred distances between secondary CM sources and cement plants are generally less than a few hundred km by road or rail[30], although longer transport distances are not uncommon (in some regions like California with poor local supply,

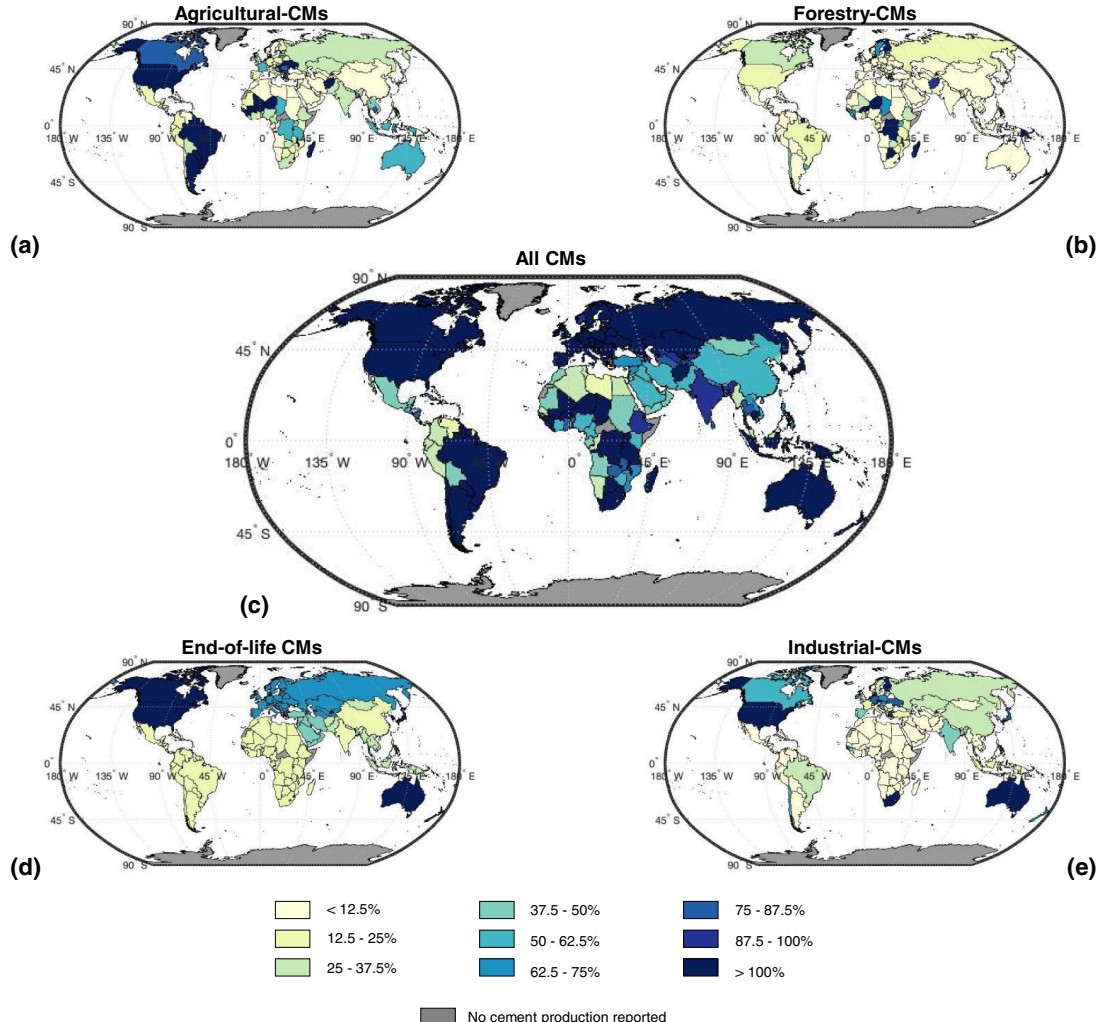

**Fig. 3 | Geographic breakdown of potential generation of secondary cementitious materials (CMs).** Data are shown on a country-by-country basis for year 2018 using units of mass potential secondary CM generation per mass cement production (i.e., Gt secondary CM/Gt cement). **a** Agricultural by-product ashes. **b** Forestry by-product ashes. **c** All secondary CMs generation considered herein. **d** End-of-life binder. **e** Industrial by-products, comprising coal combustion by-products including coal fly ash and iron and steel by-products including granulated blast furnace slag.

and for internationally traded products like cement), slightly increasing associated GHG emissions. Here, we disaggregate our data for the generation of secondary CMs globally (Fig. 2) to the country level (Fig. 3) to indicate their practical potential supply relative to demand in local/regional cement markets.

Many countries can generate secondary CMs in similar or greater cumulative quantities relative to their national cement production (Fig. 3). Almost all largest cement producers can generate secondary CMs in amounts that exceed 50% of their cement production. For example, the United States, Germany, and South Korea could have generated more industrial by-products and end-of-life binder (cumulatively) than domestic cement production in 2018. Conversely, China, Philippines, and Egypt produced more cement (1.9, 1.7, and 3.3 times respectively) than their total potential domestic generation rates of secondary CMs, so only partial clinker substitution is possible in these countries without importing. Several countries can generate more of one type of secondary CM than domestic cement production (Supplementary Information, S1); for example, agricultural by-product ashes in Brazil and United States, industrial by-products in the United States, and end-of-life binder in Canada, Japan, South Korea, and the United States. Overall, secondary CMs could substantially reduce demand for clinker in many countries, with China being a primary outlier. As the current largest producer of clinker globally (accounting for 54 mass% of global cement production in 2018[31]), China shifts the global trends. Hence while for some countries the potential to substitute clinker with secondary CMs is up to 100 mass% (i.e., completely replacing clinker in cement, Fig. 3), the theoretical global average substitution potential is up to ~86 mass% (Fig. 2). Achieving such high substitution levels will require use of alkali-activation technology.

Of the total ~3.5 Gt of secondary CMs that could have been generated globally in 2018, current utilization of secondary CMs is relatively low and mainly limited to granulated blast furnace slag and coal fly ash[30]—indicating a large untapped clinker substitution potential with other secondary CMs in several countries (Fig. 4). For example, the average European cement contains a clinker-to-cement mass ratio of ~0.75[32,33] and the average cement in the United Kingdom contains ~20 mass% coal fly ash and granulated blast furnace slag[34,35]. However, Europe produces relatively high quantities of end-of-life binder (in end-of-life concrete and mortar), most of which is currently downcycled into loose applications such as road sub-base[10,36], i.e., not recycled into cement. In general, generation of end-of-life binder (blue bars, Fig. 4) relative to cement production is high in Europe since building and infrastructure stocks are older here than in other countries. This situation presents an opportunity for the European region to greatly reduce its demand for clinker using end-of-life binder. This could be direct use (in lower quantities), as a material with similar

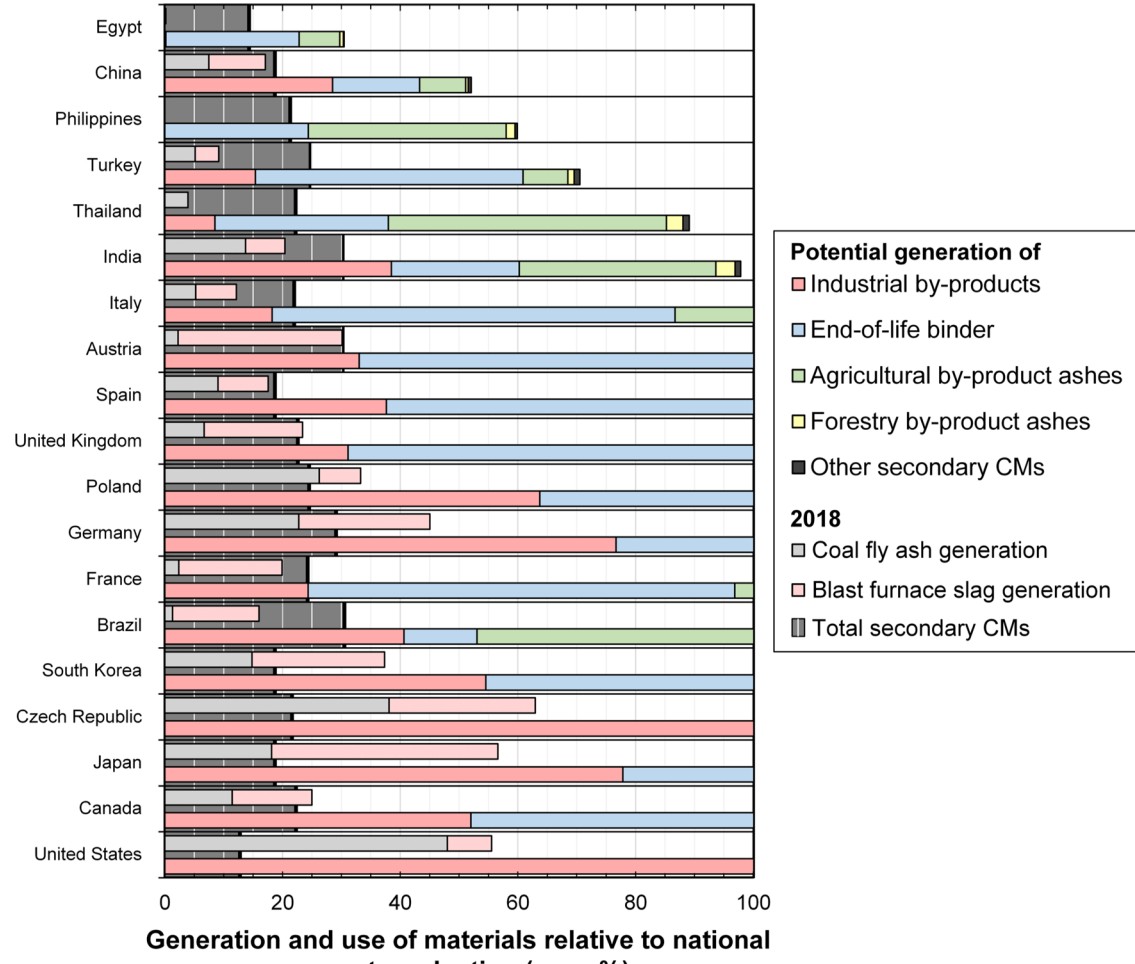

**Fig. 4 | Current (2018) vs. potential substitution of clinker for secondary cementitious materials (CMs) in several countries.** Clinker substitution potentials for Italy through the United States (as listed in Fig. 4), were greater than 100% in 2018, so a reduced number of secondary CMs are shown for these countries (see Supplementary Information S2 for the full dataset). Potential generation of secondary CMs (by CM type, lower bars, and for coal fly ash and granulated blast furnace slag, upper bars) relative to cement production in 2018 are shown for each country (colored columns). These potential generation values are comparable to the actual reported secondary CMs-to-cement ratios in 2018 (dark gray shaded backgrounds).

properties to fine limestone (Supplementary Information S1, Section S1.1.8), or as a feedstock for cement produced by carbonating end-of-life binder (in higher quantities), which can regenerate its binding capacity to become highly reactive[37]. It is thus important that technologies to efficiently separate end-of-life binder from mortar and concrete[38] and for closed loop recycling of end-of-life binder[37] are further developed and used in practice. Opportunities to increase valorization of other secondary CM types exist in other countries and regions, for example alkali-activation of coal fly ash in the United States[39], treated (e.g., calcined) bauxite residue and metallurgical slags in Europe[40], and agricultural by-product ashes in Brazil[41] (Fig. 4).

To exemplify the role that secondary CMs could play in meeting GHG emissions mitigation goals, we performed environmental life cycle assessment (LCA). Based on these results (Fig. 5), global GHG emissions from cement production could have been reduced by up to ~44% (1.3 Gt $CO_2$-eq.) by maximizing the amounts of secondary CMs utilized to substitute clinker, which is equivalent to reducing global anthropogenic GHG emissions by ~2.8% (i.e., almost equal to the total GHG emissions from Canada and Australia combined in 2018, ~2.9%[42]).

Therefore, our results show that significant reductions in $CO_2$ emissions in regional CMs cycles and their cement industries can be achieved by using locally available yet underutilized secondary CMs.

For instance, GHG emissions from cement production in Brazil could have been reduced by ~84.8% by maximizing utilization of secondary CMs, which is equivalent to reducing ~2.9% of its national GHG emissions (Fig. 5). Potential national GHG reductions were similarly high in Turkey (5.5%), South Korea (5.4%), and China (4.4%) due to the relatively larger GHG emissions coming from cement production in these countries in 2018 (i.e., 10.6%, 6.3%, and 13.3%, respectively).

High GHG emissions reduction potentials are obtained for countries that can generate similar or greater amounts of secondary CMs relative to cement production. Such countries include the United States, Canada, and the United Kingdom (Fig. 4), for which we determine cement substitution to have a potential to reduce current national GHG emissions (excluding land use, land use-change and forestry[42]) by 1–2%. For the United States, this corresponds to a reduction in the cement industry's GHG emissions of ~65 Mt $CO_2$-eq., which is approximately the total GHG emissions from Austria[42] (in 2018). On the other hand, China has a limited potential supply of secondary CMs relative to its cement production, so only relatively low clinker substitution extents can be achieved here. However, due to the massive scale of cement production in China, which was responsible for about 3.6% of global anthropogenic GHG emissions in 2018[42], clinker substitution could have theoretically avoided 548 Mt $CO_2$-eq. GHG emissions nationally. This reduction equals

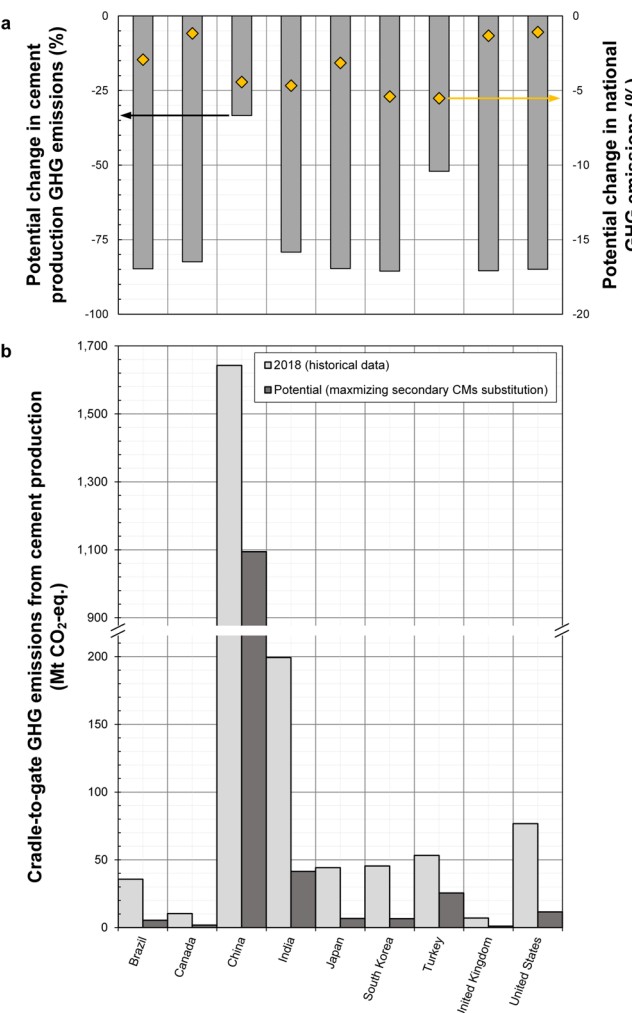

**Fig. 5 | Potential reductions in life cycle greenhouse gas (GHG) emissions by increasing clinker and cement substitution using secondary cementitious materials (CMs) in several countries. a** Potential percent reduction in GHG emissions from cement production (via increased utilization of secondary CMs, left side) and percent reduction in country-level total GHG emissions in 2018 when substituting clinker with secondary CMs (right side). **b** Current (2018) and potential (via increased utilization of secondary CMs) GHG emissions in Mt $CO_2$-eq. for cradle-to-gate cement production. Note the y-axis is cut to show GHG emissions from cement production in China and other countries using the same scale.

4.4% of China's national GHG emissions during the same year (Fig. 5), which is more than those from South Africa during 2018 (~464 Mt $CO_2$-eq.)[43].

In general, our results confirm that increased substitution of clinker, which is the most GHG intensive component of cement, with secondary CMs proportionally reduces GHG emissions. These GHG emissions reductions are only moderately offset by alkali-activation (i.e., by ~0.064 kg $CO_2$-eq. per 1.4 kg cementitious binder from the use of a sodium silicate activator in cementitious binder mixes, which is needed to achieve sufficient compressive strength development) and transport of secondary CMs (i.e., by ~0.012 kg $CO_2$-eq. per 1.4 kg cementitious binder from 150 km freight transport from the point of generation to the cement plant) (see Methods). Additionally, while the use of alkali-activation technology enables potentially large GHG emissions reductions, it would lead to trade-offs by increasing environmental impacts in other categories, notably ecotoxicity[44], the (dis) benefits of which could be substantial and need to be carefully considered.

## Discussion

Our results show that clinker and cement substitution can play a much larger role in reducing GHG emissions of cement production than it currently does, with reductions of up to 44% theoretically achievable by maximizing secondary CM utilization globally. This level of GHG emissions reduction should be viewed as an ambitious upper bound since it would require the following significant barriers to be overcome:

I. A shift in manufacturing. A massive expansion in the processing and use of a broader range of secondary CMs (e.g., agricultural by-product ashes) and alkaline activators (e.g., sodium silicate) would be needed. This will require new region-specific collection and production practices to be developed due to variability in resource supply and economic conditions.

II. Policy interventions. Standardization of various secondary CMs and secondary CM-containing materials (cements, binders, mortars, and concretes) are needed, since only a few are currently included in (inter)national cement standards[18]. These standards will most likely need to be performance-based[45] due to the wide range of feasible cement and binder formulations.

III. Investment in materials research and development. There are varying but generally low levels of technology readiness for many of the regionally scalable secondary CM-containing materials, which has persisted due to a lack of systemic research to identify appropriate pathways to their utilization. Our analysis highlights a need to better understand the properties of alkali-activated materials containing multiple secondary CMs. We suggest developing this understanding for key alkali-activated material classes (e.g., high-Ca[27], low-Ca[27], high-Fe[46], and high-Mg[47]) rather than individual mixtures, since there are many feasible mixes although the key binding phases in the classes are broadly similar.

Our study provides a comprehensive analysis of secondary CMs generation at the national level and with global coverage. In doing so, it represents a key step in overcoming this latter barrier (Figs. 3, 4). Adoption of performance-based materials standards (e.g.,[48]) as default practice, and their continued development (e.g.,[49]), will support greater use of unconventional secondary CMs, as discussed here. However, performance-based standards have been established for more than two decades without worldwide uptake in common construction practice[50], clearly demonstrating the need for further measures to achieve their application. These measures may include: (a) further industrialization of the construction sector globally[51]; (b) growth in policy support for digital construction and off-site component manufacturing; (c) use of building information modeling in design and construction, especially in developing countries where industrialization of construction generally lags behind developed countries[33,52]; (d) increased market segmentation (e.g., to facilitate matching of less conventional materials with less safety critical non-structural components); and (e) increasing business innovation in digital construction (e.g.,[53]). This is because off-site manufacturing of concrete components can greatly improve control of curing conditions, including conditions unattainable in in-situ (traditional) construction. This increased control of curing conditions has the potential to improve material quality and component performance, enabling use of unconventional materials including secondary CMs that are unsuitable for use in ready-mixed concrete. However, it requires the concrete components to rapidly develop compressive strength, which can be challenging for highly substituted composite Portland cements. There is thus a research need to investigate the properties of secondary CM-containing concretes under the wider range of feasible curing conditions in off-site manufacturing facilities relative to in-situ construction and use the resulting insights to engineer and optimize processing of by-products into secondary CMs. Another key research gap is a quantitative understanding of the systemic benefits and

impacts of digital/off-site construction systems relative to traditional in-situ construction, which although is expected to improve material efficiency and thus reduce resource impacts and GHG emissions, has not yet been rigorously studied.

Our study also highlights the potential for international trade to support increased clinker substitution levels. For example, Turkey and Egypt currently use modest amounts of secondary CMs (clinker-to-cement mass ratios of 0.75 and 0.86 in 2018, respectively) and have limited potentials to produce secondary CMs domestically (71 mass% and 30 mass% of domestic cement production in 2018, respectively), which presents an opportunity for surplus secondary CMs produced in neighboring European countries (Germany, Spain, Czech Republic, Italy, Poland, Austria—which have the potential to produce more secondary CMs than current cement production) to be used to substitute their domestic clinker production. Nonetheless, the largest cement producer, China, is not self-sufficient in locally available secondary CMs (52% of secondary CMs potentially available relative to its 2.2 Gt of cement production in 2018), and neighboring countries (e.g., India, Japan, and South Korea) would be unable to supply secondary CMs in such a massive quantity. Hence our study reinforces the need for a systematic approach to reduce $CO_2$ emissions from the CMs cycle—considering its interactions with other sectors (industry, forestry, agriculture, etc. as sources of secondary CMs), recognizing opportunities and barriers spanning multiple life cycle stages (e.g., the interplay among materials production, construction, and waste management in the digital construction paradigm), cooperating internationally (e.g., trading of secondary CMs) to increase clinker substitution globally, and improving material efficiency in the construction sector through design optimization for reduced overall demand for cementitious materials.

## Methods

### Summary

We analyzed a comprehensive selection of secondary CMs that have among the greatest potential to be commonly used in practice worldwide, including standardized[18] and less commonly used secondary CMs (e.g., forest residue ash[54]). We compiled data related to their potential generation from various sources, combining material production statistics (e.g.,[4,55–58]) and technological factors related to their upstream processing conditions (e.g., ash content[59]). We developed the latter data into by-product-to-main product ratios representative of these upstream technologies, since production data for main products from industrial (including agriculture and forestry) processes are usually reported, e.g., pig iron[4], but are not usually reported for by-products. Using the potential secondary CM generation rates at the country-level, we applied LCA to analyze the environmental impacts of cementitious binders with and without clinker substitution by secondary CMs. The LCA results, based on country-specific inventory data for major processes such as clinker production and the electricity mix, formed the basis of comprehensive GHG emission estimates for current (2018) and potential (with maximum utilization of secondary CMs) cement production. Specific statistics, methods, and assumptions used here to develop the secondary CM data and LCA models are presented in the following sub-sections.

### Coal fly ash

We estimated the total generation of fly ash, a by-product from coal-fired electricity generation, globally using International Energy Agency (IEA) data[56] and by-product-to-main product ratios for coal combustion products derived from United States (US) coal statistics[57,60]. Therefore, our global results for coal fly ash represent average US coal-fired electricity generation technology and raw materials (mainly bituminous coal), which we note can be low relative to ash generation from other nations[61]. Our results show relatively constant coal fly ash generation since 2010, at ~0.35 Gt year$^{-1}$ globally (Fig. S1,

Supplementary Information). This value lies in the expected range of coal fly ash generation (0.15–0.75 Gt[62–64]) given current rates of coal consumption for electricity generation[65], ash generation of 5–20 mass % of coal, and coal fly ash contents of 85–95 mass% of the total ash generated (with the remainder as coal bottom ash)[62].

### Flue gas desulphurisation gypsum

The main secondary CM generated from flue gas desulphurisation (FGD) is gypsum, which is a by-product produced from coal-fired electricity generation[66]. We estimated generation of FGD gypsum using the same procedure and datasets[56,57,60] as described above for coal fly ash.

### Granulated blast furnace slag

Blast furnace slag is a by-product of pig iron production. The ratio of pig iron to blast furnace slag produced depends on the iron content of the iron ore used as raw material. Typical iron concentrations in iron ore of 60–66 mass% lead to 25–30 mass% blast furnace slag per unit mass of pig iron produced[67]. We combined these data with pig iron production statistics[55] to estimate the generation of blast furnace slag globally (Fig. S3, Supplementary Information) and thus show the potential availability of granulated blast furnace slag. Therefore, our values refer to an upper limit of (ground) granulated blast furnace slag availability since they assume that all blast furnace slag is quenched (and ground) into this reactive (glassy) material.

### Silica fume

Silica fume is a very fine particulate by-product of silicon (96-99 mass%[68]) and ferrosilicon alloy (two common grades are 50 and 75 mass% silicon[69]) production. Its main source is the electric arc furnace, which is central to the production of these materials. Lesser amounts are also produced downstream of the electric arc furnace, e.g., during ladle tapping and refining. The yield of silicon (Si) from an electric arc furnace is typically 80–90 mass%, meaning that 10-20 mass% of Si in the feed is lost as silica fume in this processing step[68]. In our calculations, we assumed that Si in the feed is present as pure silica ($SiO_2$), and the reduced electric arc furnace product contains 99 mass% Si (i.e., an upper value), to determine main product-to-by-product (silica fume, $SiO_2$) ratios of 1.9–4.2 from the electric arc furnace. The amount of silica fume generated during tapping and refining varies from 7 to 13 kg $SiO_2$ per tonne Si produced[70]. We assumed that Si in the refined Si product contains 99 mass% Si, to estimate main product-to-by-product (silica fume, $SiO_2$) ratios of 78–144 from the ladle. These ranges should thus be treated as upper estimates of the amounts of silica fume that may be produced in electric arc furnaces and ladles during ferrosilicon and Si metal production. We used the median main product-to-by-product values (3.05 and 111, respectively) here, combining these results with reported historic silicon metal and ferrosilicon production statistics[71]. We further assumed no ladle slag and no losses downstream of refining (e.g., during casting) to determine our upper estimate of silica fume generation.

### Bauxite residue

Bauxite residue is a tailings type by-product from alumina production. We obtained alumina production rates using US Geological Survey (USGS) data[72], and used data from the International Aluminum Institute[73] to derive a world average main product-to-by-product (bauxite residue) ratio of 1.19. This ratio was then applied to the USGS country-level data to estimate national bauxite residue generation rates.

### Agricultural by-product ashes

We used reported agricultural crop production statistics from the Food and Agriculture Organization (FAO)[58] crop by-product-to-main product ratios[59], and ash contents of crop by-products[59], to quantify the potential availabilities of crop by-product ashes globally. Our

calculations represent upper bounds on the generation of ashes from agricultural crops, since they assume that all crop by-products are recovered and treated by energy recovery in a way that supports glassy ash generation, which will not be achieved in practice due to their various competing uses (e.g., fodder).

## Forestry by-product ashes

Similarly to agricultural by-product ashes, we used FAO statistics[58] to quantify amounts of forestry products produced globally (e.g., roundwood). The extent to which forest residue is produced and collected depends on tree species, technology used, terrain, etc. Reported values for forest residue production include 10–15 vol.% of the total above ground biomass in trees[74] and 40 vol.% of the volume of logs extracted[59]; forest residue collection rates may approach 50% (dry matter basis) of the produced amounts[75]. We applied average reported by-product-to-main product ratios (e.g., forest residue-to-roundwood)[59,74], an assumption of a 100% by-product collection rate, and average by-product ash contents (~9 mass% for forest residue)[59,76] to the FAO statistics[58], to estimate the potential generation of forestry by-product ashes globally.

## End-of-life binder

We estimated generation rates of cement removed from the built environment at end-of-life (i.e., in construction and demolition waste) globally. These rates were determined through dynamic material flow analysis modeling that captured the production rates of cementitious materials as well as their in-use periods for residential, non-residential, and civil engineering applications. The material product lifetimes as well as the per-capita saturation levels (i.e., the upper levels of per-capita demand for cementitious materials in-use) were based on values presented by Cao et al.[77] for their Medium-Moderate model. Our calculations use historical population statistics[78], and these reported values to estimate in-use cement stocks and cement removal in ten major regions of the world (North America, Europe, Africa, India, Developed Asia and Oceania, Latin America and Caribbean, Commonwealth of Independent States, Middle East, China, Other Asia)[79]. These regional statistics were disaggregated to predict national end-of-life cement generation by weighting regional production by national production statistics for year 2018 (from ref. 80).

We combined our estimates of end-of-life cement generation with typical percentages of cement end uses (15% in mortar and 80% in concrete[10]) and the means of ranges of clinker (0.21 in mortar; 0.12 in concrete), cement (0.25 in mortar; 0.18 in concrete), and binder (0.37 in mortar; 0.26 in concrete) intensities in mortar[81,82] and concrete[83] (in units of kg clinker or cement or binder per kg mortar or concrete), to determine the potential generation of end-of-life binder. Although end-of-life binder in mortar and concrete can be significantly carbonated, the extent of carbonation varies greatly depending on how the material is managed at end-of-life (extent of particle size reduction, humidity and air exposure conditions and time), and can be low for large concrete fragments or high for fine mortar and concrete particles. Therefore, we approximate the generation rate of end-of-life binder in the hydrated and uncarbonated form. We use these data to calculate that of the end-of-life binder generated in 2018, 81% was from concrete and 19% was from mortar (see Supplementary Information, Section S1.1.8). This result is an indicative estimate of the proportions of end-of-life binder generated from concrete and mortar in different countries. We expect these proportions to differ among countries, especially between developed and developing countries, although have not used country-specific data for end-of-life binder generation rates here due to poor data availability.

## Life cycle assessment

The goal of our LCA study was to quantify the maximum potential GHG emissions reductions (in $CO_2$-eq.) associated with substituting clinker for secondary CMs in selected countries. We chose 2018 as the reference year for our study, since it was the most recent year with the required data available at the time of writing. The selected countries, including the major cement producers such as China, India, Vietnam, United States, etc., represented ~70% of the global cement production in 2018. We chose a functional unit of 1.4 kg cementitious binder, which includes 0.4 kg (mixing) water and 1 kg of other binder materials (e.g., secondary CMs, clinker, gypsum, sodium silicate activator), at the concrete batching plant. This functional unit implies that the national and global average cementitious binder mixes modeled produce concrete with equivalent properties (e.g., compressive strength development). LCA results based on this functional unit were used to estimate regionalized and globally-scaled $CO_2$-eq. emissions using 2018 cement production data from the Getting the Numbers Right (GNR) database[15] and USGS cement statistics[80].

The assumption of functional equivalence between national average cementitious binder mixes is generally suitable within established mix classes (e.g., CEM I–V, LC[3]), but generally not for mixes with higher amounts of secondary CMs or those that contain significant quantities of unconventional secondary CMs (e.g., agricultural by-product ashes). Therefore, we modeled potential cementitious binders (with maximal secondary CM substitution) as sodium silicate activated materials. Sodium silicate activated materials are reported to have acceptable to desirable properties for many of the secondary CMs analyzed here (blast furnace slag, coal fly ash, calcined clays, synthetic glassy precursors, ferrous and non-ferrous metallurgical slags, bauxite residue, coal bottom ash)[84], meaning that they are broadly capable of achieving similar (or in some cases improved) compressive strength development relative to conventional cementitious binders. Cementitious materials based on agricultural and forestry by-product ashes are generally not currently used commercially, although their suitability to substitute clinker are in some cases well known (notably rice husk ash[85]). Therefore, the main source of uncertainty regarding our functional unit lies in the generally poor understanding of the properties of alkali-activated materials containing multiple secondary CMs. Such materials are theoretically feasible if (i) their bulk chemical compositions are similar to those of alkali-activated materials that have adequate properties and (ii) the secondary CMs have adequate reactivity. We show that the former is true in Supplementary Information S1 (Tables S1–S4); the latter requires additional research and is beyond the scope of this paper. In summary, our LCA results should be interpreted as indicative estimates rather than exact values due to our use of this assumption in choosing our functional unit.

The LCA study used a cradle-to-gate scope since the production life cycle stage emits the majority of GHGs from the CMs cycle[86]. The system boundaries included raw materials extraction and transport to the cement plant, pretreatment and pyroprocessing of raw meal for clinker, transport of secondary CMs to the cement plant, and transport of cement from the cement plant to the concrete batching plant (Fig. S7, Supplementary Information). Allocation of impacts to the generation of precursors to secondary CMs and their processing into secondary CMs were excluded in our LCA study; i.e., all secondary CMs were considered to carry no upstream environmental burdens. Such impacts are most pertinent in the case of economically valuable by-products, e.g., some coal fly ashes and granulated blast furnace slag[16,24,25], although their exclusion will have a limited effect on our LCA results due to the lower revenues derived from these materials than their corresponding main co-products (see the by-product-to-main-product ratios in the Supplementary Information). To assess the effect of excluding impacts from processing, we performed a sensitivity analysis whereby the waste/by-product precursors to secondary CMs were assumed to undergo the same treatment as fly ash (based on data from ref. 87). The results from this sensitivity analysis show that processing of wastes/by-products from their points of generation into secondary CMs contributes a minor amount of GHG emissions (up to

~0.012 kg $CO_2$-eq. emissions per functional unit; see Supplementary Information S1, Section S2). Additionally, avoiding allocation of these impacts facilitates consistent comparisons across materials since several of the secondary CMs considered in this work are not commercially applied. We did, however, include downstream impacts arising from: (i) 150 km of transport of secondary CMs to the cement plant (freight transport assumed on a Euro 5 type truck with a nominal payload capacity ≥32 tons), as previously reported[30,88]; (ii) another 150 kilometers of transport from the cement plant to the concrete batching plant[21]; and (iii) alkali-activation via the addition of solid sodium silicate and water based on the mix design in ref. [89] for all potential cementitious binders where maximal secondary CM substitution is used (assuming this alkali-activation step to be necessary given the significantly reduced reactivity of highly substituted cements). This activation step will likely overestimate GHG emissions since cements with non-zero clinker content do not necessarily require this much activator.

For different unit processes, emissions were based on country/region-specific inventory data (Section S2, Supplementary Information), collected from the ecoinvent database (version 3.6, cut-off system model)[90] and literature[22,24,30]. A clinker-to-gypsum mass ratio of 95:5 was assumed for all cementitious binders modeled. We scaled our inventory analysis model to achieve the functional unit (i.e., 1.4 kg of cementitious binder) with varying proportions of clinker and secondary CMs, to identify potential (i.e., maximal) $CO_2$-eq. emission savings associated with substituting clinker and cement using secondary CMs.

Life cycle impact assessment was performed using the ReCiPe 2016 midpoint impact assessment method and 100-year global warming potentials (based on the characterization factors developed by the Intergovernmental Panel on Climate Change)[91,92]. Following the impact assessment, we carried out a contribution analysis to assess the impacts of different processing steps during cement production on GHG emissions (Fig. S8, Supplementary Information).

## Data availability

The data generated in this study are provided in the Supplementary Information files. We provide two items of Supplementary Information to accompany this paper: Supplementary Information S1, a document that provides detailed information on the secondary CMs included in the analysis here and the LCA results; and Supplementary Data 1, a spreadsheet that includes the data and results presented here in tabular format.

## Code availability

The maps presented in Fig. 3 were produced in MATLAB using code from http://www.chadagreene.com to populate country borders.

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

## Acknowledgements

R.J.M. thanks Athina Preveniou and Konstantinia Papadimitriou (ADMIRIS) for discussions pertaining to bauxite residue. Funding provided by the Engineering and Physical Sciences Research Council of the U.K. (EP/S006079/1, EP/S006079/2) and Imperial College London are gratefully acknowledged. S.A.M. acknowledges funding provided by the United States National Science Foundation (CBET-2033966). D.J. acknowledges funding provided by the United States National Science Foundation (CBET-1706097). The research leading to this publication benefitted from funding received from the European Community's Horizon 2020 Program (H2020/2014-2020) under grant agreement n° 958208, EPSRC funding under grant No. EP/R010161/1, and from support from the UKCRIC Coordination Node, EPSRC grant number EP/R017727/1, which funds UKCRIC's ongoing coordination. This work represents the views of the authors, not necessarily those of the funders. Rights assertion statement: For the purpose of open access, the author has applied a Creative Commons Attribution (CC BY) license to any Author Accepted Manuscript version arising.

## Author contributions

R.J.M. originated the idea, led research design and write-up of the paper, and contributed to research execution; I.H.S. contributed to research design, write-up, and research execution; D.J. contributed to research design and write-up; and S.A.M. contributed to research design, research execution, and write-up.

## Competing interests

The authors declare no competing interests.
