## [Peer Review File · Nature Communications]

REVIEWER COMMENTS

Reviewer #1 (Remarks to the Author):

The purpose of this paper is good – mapping the availability of CMs to local cement production, and thus determining the potential for replacement and nominal reduction in CO₂ emissions. The execution is less so. I found the paper rather difficult to follow, not well organised, and the narrative is not clear. It needs a major revision, and some key data elements need to be extracted from the LCA and put into the main body of the paper before it can be reviewed properly. In essence, the content that deals with the distribution and availability of CMs is very good, but the subsequent assertions about how these could reduce the CO₂ emissions of the cement industry are unfounded as-written, although I'm sure the relevant data is buried within the depths of the LCA. Either firm up these assertions with practical data on eCO₂ for CMs plus consideration of practical replacement rates, or simply stick to reporting the CM availability. I think there is the germ of a very good and useful paper here, but it has been rushed. This review appears harsh, but if you want this work to get the impact that it deserves, then you need to make it bullet-proof by addressing these issues, or it will simply be dismissed. Address these issues, and you could have a landmark paper on your hands.

29-31: One cannot 100% substitute cement clinkers with any of the CMs discussed here – it is chemically impossible. This assertion should be removed. I get that you say 'could have', but this implies 'should have' to a reader not versed in the minutiae of cement chemistry and is thus misleading. Alternatively, carefully define what you mean by 'substitution'. If you mean 'use a material other than Portland Cement concrete, with or without CMs', make that clear. If so, also make it clear what the CO₂ emissions are associated with the alternative product. Cement and concrete have, by mass or volume and very often by functional unit, the lowest eCO₂ values of any material; any replacement material must clearly and demonstrably have lower eCO₂ per functional unit.

52 et seq: The distinction between clinker, binder and cement is not made clear enough here. I'm an expert and I had to read it several times. You will need to explain this, probably with a diagram. The lay reader would currently find it very difficult to understand the difference between clinker, cement and binder. Figure 1 does not help at all – I would simplify it or remove it. Substitution at the clinker level implies using CMs as raw materials i.e. feeding them into the kiln, and is controlled by the cement producer and would have minimal carbon impact. Substitution at the cement level involves pre-blending CMs with cement (i.e. ground clinker) prior to it being marketed as a binder, and is thus also controlled by the cement producer. Substitution at the binder level involves the cement customer (i.e. contractor or increasingly, ready-mix concrete supplier) replacing some of the cement supplied to her with CM while mixing concrete. This would be a clearer distinction. Also, I see no reason to distinguish between mortar and concrete here – it just confuses the issue. A footnote or parenthetical note explain that mortar is a subset of concrete early on would be sufficient.

73: 100% "replacement" is not possible even at this scale. The papers to which you refer concern so-called geopolymers; effectively a completely different material which must be activated with extremely strong alkali (the production and use of which have severe environmental and health impacts) and cured under carefully controlled conditions, often involving heat. This should be removed or reworded.

96 et seq: the shift from absolute to relative numbers when discussing cement and CMs respectively is not helpful here. Has the generation of CMs remained stable but cement production increased? Please report the absolute (tonnages) here so people can see the full picture. Also, is the fall from 75% to 67% a result of your analysis or someone else's? You introduce the concept of primary CMs here (i.e. limestone, clays [MK]) – this should be introduced much earlier, before the results section, as it adds to the confusion here since you do not include limestone/MK in your onward analysis. This is why the clinker-cement-binder distinction must be made crystal clear early on.

112: See above – you suddenly jump from talking about secondary CM in the figure to primary CMs. You should deal with primary CMs in the introduction to avoid confusing the reader.

125: You should make it clear what you mean by 'allocated'. Most waste-derived CMs are only 'zero carbon' because their price is such that they generate less than the 1% threshold of the revenue associated with the parent product that many LCA practitioners observe. As demand and thus price for CMs increases, many of them will be allocated a significant share of the carbon associated with the primary process; they will no longer be nominally 'zero carbon'. Steel and electricity producers would love to allocate some carbon to GGBS and PFA respectively and sooner or later this will happen, and other CMs will follow suit. This has been discussed in the open literature and you should acknowledge that.

129: I'm afraid without an explanation of replacement rates, you can't really say this. Use of 'multiple concurrent secondary CMs' is simply not possible; the chemistry of the system doesn't allow it. The replacement rate is limited by the active silica and calcium content of the CM. Many of these contain no calcium and thus a replacement rate of >40% is not possible; the concrete will not harden, or will harden too slowly for practical purposes. While I appreciate that you are calculating upper bounds here (and you make this clear later in the paper), without directly looking at the effect of replacement rates no one will take this seriously, which would be a shame as I think the basic research is very useful.

136: Not so. The US is importing PFA from India as we speak, which means it has a price, which means it should have a carbon allocation.

153 et seq: Again, 100% substitution is impossible. Using CMs plus alkali to make geopolymers is not proven to be lower carbon than using normal concrete, despite what its adherents profess, and it simply cannot be used in the same way as normal concrete. Many of the agricultural ashes are unproven in concrete outside of the lab.

176 et seq: I'm very unclear about how you propose that end-of-use binder should be used as a cement replacement. It simply won't work, unless you are proposing to use it as kiln feedstock, in which case from a carbon perspective there's no point. Can you clarify this? What exactly do you propose that people do with end-of-life binder to make it a substitute/replacement? Ref 30 employs a heat treatment of up to 1100°C – hardly low carbon, they are basically looking at DTA residues i.e. 20 mg at a time – and ref 31 only reports chemical activity, not associated strength development.

186: Bauxite residue must be calcined before use at 600-800°C. Even then, the replacement rates achievable are low.

198: Did your LCA take account of the achievable replacement rates, or assume that 100% replacement was possible in every case? What eCO₂ values were allocated to each of the CMs? I appreciate that this may be buried in the LCA but these need to be in the main body of the paper to aid transparency. Also, what variability have you assigned to these values? The eCO₂ of all of these CMs is not well established and reported values can vary by an order of magnitude. Everything following this cannot be reviewed without this data.

Reviewer #2 (Remarks to the Author):

The paper is very interesting and deal with the potential of SCMs as clinker replacement considering two scenarios and regional data, very hard to find.

I appreciated the organization of supplementary materials both the pdf and the excel file.
Main comments/suggestions.

1. In my opinion authors should be think about changing the title. Although in the paper the end-of-life binder is considered, "recycling cementitious materials" can lead readers thinking only about concrete recycling.
2. Line 101 – please cite the source of the information
3. Authors describe the possibility of use alternative binders with low (or not) Portland clinker and an

activator. In my opinion considerations about the availability/mass balance of the activator(s) should be discussed.

4. Authors also propose the off-site manufacturing as a solution to improve the diffusion of the alternative binder, this because the controlled production conditions. They should be mentioned that doing this means that alternative binders need to have early high strength. On the other side, off site manufacturing is not so diffuse in developing Countries that are the larger cement consumers.

5. Line 453-454. It is a huge simplification. Authors know that SCMs have usually high specific surface area, and this implies that water demand increase and so the porosity of materials and the strength decreases. Please comment about this in the text.

6. About the end-of-life binders. The percentage distribution in 2018 gives 18% mortar and 81% concrete for each considered Country. Maybe this distribution is ok for developed Countries... Please check and comment.

7. Line 474 – 477. The sentence is not clear. Authors mean that when the cement contains low amount of clinker Portland, do they suggest adding the activator?

8. Line 480. Authors used ecoinvent database. They should comment about the limitation of source of data, especially related to developing Countries.

9. Main comment for the Supp. Information. My suggestion is to try adjusting the Y axis scale of charts cutting the axis in 1 to show the 4 or try to consider a log scale for Y axis to make chart more readable.

We thank the Reviewers for their comments and hope that our responses and revisions to the manuscript satisfy their concerns. Page and line numbers refer to the revised version of the manuscript without markup. Reviewer comments are reproduced here verbatim in blue. Our responses (black text) are presented alongside each of the specific reviewer comments (blue text).

Reviewer Comments:

NCOMMS-22-07440-T: Recycling cementitious materials can reduce annual greenhouse gas emissions by 1.4 gigatons

Reviewer #1: General comments:

The purpose of this paper is good – mapping the availability of CMs to local cement production, and thus determining the potential for replacement and nominal reduction in CO₂ emissions. The execution is less so. I found the paper rather difficult to follow, not well organised, and the narrative is not clear. It needs a major revision, and some key data elements need to be extracted from the LCA and put into the main body of the paper before it can be reviewed properly. In essence, the content that deals with the distribution and availability of CMs is very good, but the subsequent assertions about how these could reduce the CO₂ emissions of the cement industry are unfounded as-written, although I’m sure the relevant data is buried within the depths of the LCA. Either firm up these assertions with practical data on eCO₂ for CMs plus consideration of practical replacement rates, or simply stick to reporting the CM availability. I think there is the germ of a very good and useful paper here, but it has been rushed. This review appears harsh, but if you want this work to get the impact that it deserves, then you need to make it bullet-proof by addressing these issues, or it will simply be dismissed. Address these issues, and you could have a landmark paper on your hands.

We appreciate Reviewer 1’s detailed and constructive comments. We agree that this can be a landmark paper.

No.	Specific Reviewer comment	Author’s response
1.	29-31: One cannot 100% substitute cement clinkers with any of the CMs discussed here – it is chemically impossible. This assertion should be removed. I get that you say ‘could have’, but this implies ‘should have’ to a reader not versed in the minutiae of cement chemistry and is thus misleading. Alternatively, carefully define what you mean by ‘substitution’. If you mean ‘use a material other than Portland Cement concrete, with or without CMs’, make that clear. If so, also make it clear what the CO₂ emissions are associated with the alternative product. Cement and concrete have, by mass or volume and very often by functional unit, the lowest eCO₂ values of any material; any replacement material must clearly and	We have clarified that these secondary materials could have been used to partially substitute Portland cement clinker and in alkali-activated materials by adjusting the text on lines 29 and 33-35, following the suggestions by the Reviewer. We have also clarified that we use the term ‘substitution’ to refer to both complete and partial substitution on line 56. Various edits have been made throughout the manuscript to make it clearer that alkali-activation is included here as a type of cement substitution. This technology can achieve 100% substitution of Portland cement (and thus clinker).

	demonstrably have lower eCO ₂ per functional unit.	
2.	52 et seq: The distinction between clinker, binder and cement is not made clear enough here. I'm an expert and I had to read it several times. You will need to explain this, probably with a diagram. The lay reader would currently find it very difficult to understand the difference between clinker, cement and binder. Figure 1 does not help at all – I would simplify it or remove it. Substitution at the clinker level implies using CMs as raw materials i.e. feeding them into the kiln, and is controlled by the cement producer and would have minimal carbon impact. Substitution at the cement level involves pre-blending CMs with cement (i.e. ground clinker) prior to it being marketed as a binder, and is thus also controlled by the cement producer. Substitution at the binder level involves the cement customer (i.e. contractor or increasingly, ready-mix concrete supplier) replacing some of the cement supplied to her with CM while mixing concrete. This would be a clearer distinction. Also, I see no reason to distinguish between mortar and concrete here – it just confuses the issue. A footnote or parenthetical note explain that mortar is a subset of concrete early on would be sufficient.	We have provided clearer definitions of clinker and binder, and clarified the distinction between these materials and mortar and concrete on lines 43-47 and 59-60. We hope the Reviewer finds the text satisfactorily clear with these changes implemented. We have simplified Figure 1 to more clearly show that the secondary CMs are used in the binder. We prefer to explicitly refer to mortar in the main text since it is a significant end use of cement.
3.	73: 100% “replacement” is not possible even at this scale. The papers to which you refer concern so-called geopolymers; effectively a completely different material which must be activated with extremely strong alkali (the production and use of which have severe environmental and health impacts) and cured under carefully controlled conditions, often involving heat. This should be removed or reworded.	Determining the plausible substitution levels of cementitious materials at the macroscale is the subject of this paper – we motivate the need for this in this paragraph. Our findings (Fig. 2) support the Reviewer’s comments that 100% substitution of cement for secondary cementitious materials is not possible at the global scale. This is not known to the reader at this stage of the manuscript, and we explicitly note that the potential is ‘unclear’ here. Our definition of substitution includes substitution of cementitious binder, and geopolymers are one example of this. We have made changes throughout the manuscript to make this clearer. We have included additional text on lines 274-277 noting the broader trade-offs in environmental impacts involved in Portland cement for alkali-activated material substitution, which we agree are important.
4.	96 et seq: the shift from absolute to relative numbers when discussing cement and CMs respectively is not helpful here. Has the generation of CMs remained stable but cement production increased? Please report the absolute (tonnages) here so people can see the full picture. Also, is the fall from 75% to 67% a result of your analysis or someone else’s?	We now provide absolute masses in all instances where % shares are shown. Both secondary CM generation and cement production have increased during 2002-2018 (the latter increased faster). The fall in secondary CMs share is based on our own analysis – this has been recalculated and revised.

	You introduce the concept of primary CMs here (i.e. limestone, clays [MK]) – this should be introduced much earlier, before the results section, as it adds to the confusion here since you do not include limestone/MK in your onward analysis. This is why the clinker-cement-binder distinction must be made crystal clear early on.	We have revised the introduction to clarify the terms clinker, cement, and binder, on lines 43-47 and 59-60. We have also briefly explained primary CMs on lines 82-85. We hope the Reviewer finds the text satisfactorily clear with these changes implemented.
5.	112: See above – you suddenly jump from talking about secondary CM in the figure to primary CMs. You should deal with primary CMs in the introduction to avoid confusing the reader.	We have added text on lines 82-85 in the introduction that introduces primary CMs, following the Reviewer’s suggestion.
6.	125: You should make it clear what you mean by ‘allocated’. Most waste-derived CMs are only ‘zero carbon’ because their price is such that they generate less than the 1% threshold of the revenue associated with the parent product that many LCA practitioners observe. As demand and thus price for CMs increases, many of them will be allocated a significant share of the carbon associated with the primary process; they will no longer be nominally ‘zero carbon’. Steel and electricity producers would love to allocate some carbon to GGBS and PFA respectively and sooner or later this will happen, and other CMs will follow suit. This has been discussed in the open literature and you should acknowledge that.	We agree that some secondary CMs are significantly valuable and can be allocated non-zero shares of environmental impacts. For simplicity, and since this effect is minor relative to the reduction of clinker content ¹ , we assume that they are allocated zero impacts. We have clarified and acknowledged this (including the uncertainty that it introduces) on lines 135-137 and 535-544, and in the Supplementary Information document on lines 409-413.
7.	129: I’m afraid without an explanation of replacement rates, you can’t really say this. Use of ‘multiple concurrent secondary CMs’ is simply not possible; the chemistry of the system doesn’t allow it. The replacement rate is limited by the active silica and calcium content of the CM. Many of these contain no calcium and thus a replacement rate of >40% is not possible; the concrete will not harden, or will harden too slowly for practical purposes. While I appreciate that you are calculating upper bounds here (and you make this clear later in the paper), without directly looking at the effect of replacement rates no one will take this seriously, which would be a shame as I think the basic research is very useful.	We have added text on lines 146-151, 299-304, and 507-528, and in the Supplementary Information document on lines 331-367 to provide this explanation of the types of materials that can be used to achieve the replacement rates discussed in the paper, and the implications of our assumptions. In the Supplementary Information document, we show that the bulk oxide compositions of the national average cementitious binder mixes that we model in our LCA study are similar to those for typical alkali-activated material solid precursors (metakaolin, coal fly ash, blast furnace slag). The results suggest that these national average cementitious binder mixes are feasible for alkali activation, provided they are sufficiently reactive. We comment on this in the main text.
8.	136: Not so. The US is importing PFA from India as we speak, which means it has a price, which means it should have a carbon allocation.	We believe that our statement is correct: “Therefore, distances between secondary CM sources and cement plants are generally constrained to less than a few hundred km by road or rail.” However, we have noted that international transporting of CMs exists, and also longer transport distances for some regions like California, in line with the Reviewer’s comments, and added text to discuss this factor on lines 156-160, and in the Supplementary Information document on lines 406-409. We used existing LCA studies to set the transport distance of 150 km ² .

9.	153 et seq: Again, 100% substitution is impossible. Using CMs plus alkali to make geopolymers is not proven to be lower carbon than using normal concrete, despite what its adherents profess, and it simply cannot be used in the same way as normal concrete. Many of the agricultural ashes are unproven in concrete outside of the lab.	We have included text stating that alkali-activation technology will be needed to achieve the high substitution levels discussed (line 191). We disagree with the Reviewer’s opinion that geopolymer concrete is not ‘lower carbon’ than normal concrete. It has been established through at least several studies^{1,3,12,13,4-11} that alkali-activated materials generally have reduced (in some cases very significantly) climate change impacts relative to conventional (composite) Portland cements. The reduction of clinker content in cement is the most important factor here. The appropriate choice of functional unit and mix design also play major roles in the results. Allocation (of impacts to secondary CMs such as fly ash) plays a lesser role. We take this literature into account in our LCA study. We include discussion of the suitability of the secondary CMs analyzed here to substitute clinker on lines 507-528.
10.	176 et seq: I’m very unclear about how you propose that end-of-use binder should be used as a cement replacement. It simply won’t work, unless you are proposing to use it as kiln feedstock, in which case from a carbon perspective there’s no point. Can you clarify this? What exactly do you propose that people do with end-of-life binder to make it a substitute/replacement? Ref 30 employs a heat treatment of up to 1100°C – hardly low carbon, they are basically looking at DTA residues i.e. 20 mg at a time – and ref 31 only reports chemical activity, not associated strength development.	We have included text to describe how end-of-life binder can be used as a secondary CM on lines 203-208. In the Supplementary Information document we include detailed discussion of the different secondary CMs studied here, and end-of-life binder is discussed in Section S1.8. We have included a reference to this section in the added text.
11.	198: Did your LCA take account of the achievable replacement rates, or assume that 100% replacement was possible in every case? What eCO₂ values were allocated to each of the CMs? I appreciate that this may be buried in the LCA but these need to be in the main body of the paper to aid transparency. Also, what variability have you assigned to these values? The eCO₂ of all of these CMs is not well established and reported values can vary by an order of magnitude. Everything following this cannot be reviewed without this data.	We assumed maximal replacement rates for each country (in line with secondary CM generation rates, determined here). For example: (1) Since the U.S. had more than 100% potential secondary CMs available, we modelled 100% of the clinker to be replaced; (2) China had the potential to generate 52% secondary CMs relative to cement production in 2018, thus only 52% of clinker was replaced in this case. We now explicitly show and discuss the replacement rates used in the LCA models in the Supplementary Information document on lines 331-367 including Tables S1-S4. We modelled secondary CMs as burden free from the point of generation, i.e. freight transport of 150 km is included here. This has been explained in the main text on lines 272-274 and 544-551, and in the Supplementary Information document on lines 406-413. We agree that CO₂-eq. emissions values for many of secondary CMs remain poorly reported, but we consider this to be outside the scope of this paper.

		We have added text discussing the implications of this allocation issue on lines 134-137 and in the Supplementary Information document on lines 409-413.
12.	186: Bauxite residue must be calcined before use at 600-800°C. Even then, the replacement rates achievable are low.	We agree that bauxite residue must be treated before it can be utilized as an effective clinker substitute. Calcination is one option. We have included text on line 212 “(treated, e.g. calcined)” accordingly. Our data (Fig. 2, Supplementary Information spreadsheet) show that bauxite residue generation is low compared to cement production, however, once treated it can still be used to substitute clinker in cement.

Reviewer #2: General comments:

The paper is very interesting and deal with the potential of SCMs as clinker replacement considering two scenarios and regional data, very hard to find. I appreciated the organization of supplementary materials both the pdf and the excel file. Main comments/suggestions:

No.	Specific Reviewer’s comment	Author’s response
13.	In my opinion authors should be think about changing the title. Although in the paper the end-of-life binder is considered, “recycling cementitious materials” can lead readers thinking only about concrete recycling.	We have changed the title to “Cement substitution with secondary materials can reduce annual global CO ₂ emissions by up to 1.3 gigatons” to avoid this confusion.
14.	Line 101 – please cite the source of the information	We have used our own analysis to derive these numbers. The data comes from sources mentioned in the Methods Section and given in the Supplementary Information spreadsheet, summarized in Fig. 2. We now include an explicit citation to Fig. 2 in the text here (line 111).
15.	Authors describe the possibility of use alternative binders with low (or not) Portland clinker and an activator. In my opinion considerations about the availability/mass balance of the activator(s) should be discussed.	We have now added details about the availability of activators on lines 151-154.
16.	Authors also propose the off-site manufacturing as a solution to improve the diffusion of the alternative binder, this because the controlled production conditions. They should be mentioned that doing this means that alternative binders need to have early high strength. On the other side, off site manufacturing is not so diffuse in developing Countries that are the larger cement consumers.	We agree with the Reviewer. Text has been added on lines 315-317 and 324-326 to address these points.
17.	Line 453-454. It is a huge simplification. Authors know that SCMs have usually high specific surface area, and this implies that water demand increase and so the porosity of materials and the strength decreases. Please comment about this in the text.	We agree that is a significant assumption, although we believe it is appropriate here since it is important in enabling us to determine the upper potential of cementitious substitution with secondary CMs to reduce climate change impacts.

		We have revised the text on lines 500-502 and added a paragraph on lines 507-528 to clarify and discuss this assumption, justify its use here, and discuss its implications for readers interpreting our results.
18.	About the end-of-life binders. The percentage distribution in 2018 gives 18% mortar and 81% concrete for each considered Country. Maybe this distribution is ok for developed Countries... Please check and comment.	The Reviewer is correct (although the percentages are 19% mortar and 81% concrete). We have added text to discuss this on lines 485-491.
19.	Line 474 – 477. The sentence is not clear. Authors mean that when the cement contains low amount of clinker Portland, do they suggest adding the activator?	Yes, the Reviewer is correct: We intend to communicate that the amount of activator modelled in this study is more than what we expect will be required for alkali-activated materials containing some Portland cement clinker – thus slightly overestimating GHG emissions. This is to help readers interpret our results accordingly. We have revised the text to make this clearer (lines 551-553).
20.	Line 480. Authors used ecoinvent database. They should comment about the limitation of source of data, especially related to developing Countries.	We have revised the text to highlight this important comment/suggestion in the Supplementary Information document on lines 413-416.
21.	Main comment for the Supp. Information. My suggestion is to try adjusting the Y axis scale of charts cutting the axis in 1 to show the 4 or try to consider a log scale for Y axis to make chart more readable.	We purposely plotted these data in the relevant range for Portland cement production (up to 4 Gt) to show that the generate rates of secondary CMs are small individually, and we note this in the figure captions. Our preference is to keep the figures plotted with the current y-axis scales.

We have additionally made minor editorial changes to the text throughout the manuscript to improve clarity. The authors appreciate the constructive comments of the reviewers in helping to improve the manuscript, and we hope that the changes made address the relevant concerns.

References

1. Habert, G. & Ouellet-Plamondon, C. Recent update on the environmental impact of geopolymers. *RILEM Tech. Lett.* **1**, 17 (2016).
2. Miller, S. A., John, V. M., Pacca, S. A. & Horvath, A. Carbon dioxide reduction potential in the global cement industry by 2050. *Cem. Concr. Res.* **114**, 115–124 (2018).
3. Jamieson, E., McLellan, B., Van Riessen, A. & Nikraz, H. Comparison of embodied energies of Ordinary Portland Cement with Bayer-derived geopolymer products. *J. Clean. Prod.* **99**, 112–118 (2015).
4. Turner, L. K. & Collins, F. G. Carbon dioxide equivalent (CO₂-e) emissions: A comparison between geopolymer and OPC cement concrete. *Constr. Build. Mater.* **43**, 125–130 (2013).

5. Habert, G., D'Espinose De Lacaillerie, J. B. & Roussel, N. An environmental evaluation of geopolymer based concrete production: Reviewing current research trends. *J. Clean. Prod.* **19**, 1229–1238 (2011).
6. McLellan, B. C., Williams, R. P., Lay, J., Van Riessen, A. & Corder, G. D. Costs and carbon emissions for geopolymer pastes in comparison to ordinary portland cement. *J. Clean. Prod.* **19**, 1080–1090 (2011).
7. Fawer, M., Concannon, M. & Rieber, W. LCA case studies LCI for the production of sodium silicate life cycle inventories for the production of sodium silicates. *LCA Case Stud.* **4**, 207–212 (1999).
8. Ouellet-Plamondon, C. & Habert, G. Life cycle assessment (LCA) of alkali-activated cements and concretes. *Handbook of Alkali-Activated Cem. Mortars Concr.* 663–686 (2015) doi:10.1533/9781782422884.5.663.
9. Ouellet-Plamondon, C. M. & Habert, G. Self-compacted clay based concrete (SCCC): Proof-of-concept. *J. Clean. Prod.* **117**, 160–168 (2016).
10. Teh, S. H., Wiedmann, T., Castel, A. & de Burgh, J. Hybrid life cycle assessment of greenhouse gas emissions from cement, concrete and geopolymer concrete in Australia. *J. Clean. Prod.* **152**, 312–320 (2017).
11. Salas, D. A., Ramirez, A. D., Ulloa, N., Baykara, H. & Boero, A. J. Life cycle assessment of geopolymer concrete. *Constr. Build. Mater.* **190**, 170–177 (2018).
12. Teh, S. H., Wiedmann, T. & Moore, S. Mixed-unit hybrid life cycle assessment applied to the recycling of construction materials. *J. Econ. Struct.* **7**, 1–25 (2018).
13. Yao, Y., Hu, M., Di Maio, F. & Cucurachi, S. Life cycle assessment of 3D printing geo-polymer concrete: An ex-ante study. *J. Ind. Ecol.* **24**, 116–127 (2020).

REVIEWERS' COMMENTS

Reviewer #1 (Remarks to the Author):

I'm happy that the authors have addressed or rebuffed the majority of my comments. I still disagree that geopolymers are the answer, and that their lower carbon credentials have been conclusively proved, and that the current tacit assumption that wastes from high-energy processes are 'zero-carbon', but this is at odds with the prevailing wind and shouldn't detract from the paper. I suggest it is now published.

Reviewer #2 (Remarks to the Author):

The paper has been improved a lot; it is clearer and more readable.

Below few comments/suggestions

Main text

Line 154. You should comment that the NaOH is also used in many other industries, so the availability could always be low.

Suppl. Mat

Line 275-277. I did not see in the text any comment related to the impact of processing (comminution/separation) the end-of-life cementitious materials (concrete or mortar) to obtain the end-of-life hydrated cement paste. Did you consider it?

Line 312-313. According to the definition of the "cementitious binder" in the main text, 1kg of cementitious binder = 400ml water + 600 g powder

Line 321-323. 926 g solid + 74g solid sodium silicate + 400 ml water = 1.4 kg; please check, and correct/clarify

Line 359. Bituminous

We thank the Reviewers for their comments and hope that our final revisions to the manuscript satisfy their comments. Page and line numbers refer to the revised version of the manuscript without markup. Reviewer comments are reproduced here verbatim in blue. Our responses (black text) are presented alongside each of the specific reviewer comments (blue text).

Reviewer Comments:

NCOMMS-22-07440-T: Cement substitution with secondary materials can reduce annual global CO₂ emissions by up to 1.3 gigatons

Reviewer #1:

No.	Specific Reviewer comment	Author's response
1.	I'm happy that the authors have addressed or rebuffed the majority of my comments. I still disagree that geopolymers are the answer, and that their lower carbon credentials have been conclusively proved, and that the current tacit assumption that wastes from high-energy processes are 'zero-carbon', but this is at odds with the prevailing wind and shouldn't detract from the paper. I suggest it is now published.	We thank the reviewer for their valuable input based on which several modifications/revisions were made thus significantly improving the quality of this paper. As no changes in the manuscript are suggested in this review round, this comment is received positively and appreciated without further modification to the manuscript.

Reviewer #2: The paper has been improved a lot; it is clearer and more readable. Below few comments/suggestions

No.	Specific Reviewer's comment	Author's response
1.	Main text Line 154. You should comment that the NaOH is also used in many other industries, so the availability could always be low.	We have added new text addressing this comment on line 151-153.
2.	Suppl. Mat Line 275-277. I did not see in the text any comment related to the impact of processing (comminution/separation) the end-of-life cementitious materials (concrete or mortar) to obtain the end-of-life hydrated cement paste. Did you consider it?	These impacts are not included in our LCA results. To address this comment, we have now included a sensitivity analysis (in Supplementary Information S1) showing the effect of including treatment impacts on our LCA results, finding that the effect is minor. We have also included some text summarizing this in the main text on lines 544-550.
3.	Line 312-313. According to the definition of the "cementitious binder" in the main text, 1kg of cementitious binder = 400ml water + 600 g powder Line 321-323. 926 g solid + 74g solid sodium silicate + 400 ml water = 1.4 kg; please check, and correct/clarify	We thank the reviewer for highlighting this issue. We have made necessary corrections throughout the manuscript (e.g. lines 501-504) and Supplementary Information S1 document to better and clearly describe the functional unit, which is 1.4 kg cementitious binder.

4.	Line 359. Bituminous	We have corrected the spelling error, as suggested.
----	----------------------	---

Additionally, we have made some minor editorial changes throughout the manuscript to improve readability.